# Cytotoxicity of Deoxynivalenol after Being Exposed to Gaseous Ozone

**DOI:** 10.3390/toxins11110639

**Published:** 2019-11-02

**Authors:** Dongliang Ren, Enjie Diao, Hanxue Hou, Peng Xie, Ruifeng Mao, Haizhou Dong, Shiquan Qian

**Affiliations:** 1Jiangsu Key Laboratory for Food Safety & Nutrition Function Evaluation, Huaiyin Normal University, Huai’an 223300, China; mg0701042@stu.hytc.edu.cn (D.R.); 8201611034@hytc.edu.cn (P.X.); 8201711047@hytc.edu.cn (R.M.); sqqian@hytc.edu.cn (S.Q.); 2College of Food Science & Engineering, Shandong Agricultural University, Tai’an 271018, China; hzhdong@sdau.edu.cn; 3Jiangsu Collaborative Innovation Center of Regional Modern Agriculture & Environmental Protection, Huaiyin Normal University, Huai’an 223300, China

**Keywords:** deoxynivalenol (DON), cytotoxicity, MTT assay, IC_50_, ozone detoxification

## Abstract

In this study, deoxynivalenol (DON) in aqueous solution was exposed to gaseous ozone for periods ranging from 0 to 20 min. The degradation efficiency and cytotoxicity of DON were investigated after being treated by ozone. The results showed that DON was rapidly degraded from 10.76 ± 0.09 mg/L to 0.22 ± 0.04 mg/L within 15 min (*P* < 0.05), representing a reduction of 97.95%, and no DON was detected after being exposed to 14.50 mg/L of ozone at a flow rate of 80 mL/min for 20 min. The degradation of DON depended on the ozone exposure time, and followed the first-order kinetic model (*R*^2^ = 0.9972). Human hepatic carcinoma (HepG2) and Henrietta Lacks (Hela) cells were used to evaluate the cytotoxicity of DON treated by ozone using the 3-(4,5-Dimethyl-2-thiazolyl)-2,5-diphenyl-2-H-tetrazolium bromide (MTT) assay. The half-maximal inhibitory concentrations (IC_50_) values of DON on HepG2 and Hela cells were 2.10 and 1.33 mg/L after 48 h of exposure, respectively, and showed a dose-dependent manner. The cell vitalities of HepG2 and Hela cells on DON were both evidently improved after being exposed to ozone for 15 min, and there were no significant differences between the negative control and that treated at 20 min of ozone exposure. Gaseous ozone can potentially be used as a new method to detoxify DON in agricultural products.

## 1. Introduction

Deoxynivalenol (DON) is one of the most prevalent type B trichothecenes, and is mainly produced by *Fusarium graminearum* and *Fusarium culmorum* [1]. It is usually found in many agricultural products, especially in wheat and its products [2,3]. It can have serious toxic effects on animals and humans by changing the functions and viability of cells [4]. DON ingested at higher doses causes nausea, vomiting, and weight loss, and at lower doses, food refusal [5]. DON has been verified as exhibiting immunotoxicity, genotoxicity, carcinogenicity, cytotoxicity, and chronic-toxicity through inducing oxidative damage and inhibiting the synthesis of protein, RNA, and DNA, as well as chromosome aberration, in experimental animals [6,7]. It poses a potential health risk to infants, toddlers, and other children, according to the estimated mean chronic dietary exposure to DON above the group tolerable daily intake of 1 μg/kg bw per day, and also to adolescents and adults at high exposure [8]. Based on the toxicity of DON, many physical, chemical, and biological methods have been used to remove or degrade it in agricultural products [9,10,11]. Among these methods, in recent years, ozone has been widely used to degrade DON in cereals and cereal-based products due to its strong detoxification capacity, high safety, low cost, and ability to remove DON without leaving toxic residues [12,13]. However, the safety of DON after detoxification by ozone is a concern of consumers. Therefore, the aim of this present work is to investigate the ozonolysis efficiency of DON, and to evaluate the cytotoxicity of DON after being exposed to gaseous ozone by the 3-(4,5-Dimethyl-2-thiazolyl)-2,5-diphenyl-2-H-tetrazolium bromide (MTT) assay using human hepatic carcinoma (HepG2) cells and Henrietta Lacks (Hela) cells (an aggressive glandular cervical cancer cell). Additionally, the half-maximal inhibitory concentrations (IC_50_) of DON on the HepG2 and Hela cells were also determined.

## 2. Results

### 2.1. Change of DON during Ozone Exposure

The changing concentration of DON during the process of ozone exposure is shown in Figure 1. As seen from Figure 1, the concentration of DON significantly decreased with the increase of ozone exposure times, in the range of 0 to 20 min (*p* < 0.05). It was decreased from 10.76 ± 0.09 mg/L (for the control without ozone exposure) to 3.02 ± 0.09 mg/L (for the one exposed to 14.50 mg/L of ozone for 5 min at an 80 mL/min flow rate), which was a 71.93% reduction. After 15 min of ozone exposure, the residual DON concentration was only 0.22 ± 0.04 mg/L, corresponding to a reduction of 97.95%, and DON was completely removed after 20 min of ozone exposure. The reduction of DON depends on the ozone exposure time, which follows a first-order kinetic model (*y* = 39.866 *e*^–1.281*x*^) with a correlation coefficient (*R*^2^) of 0.9972.

### 2.2. Cytotoxicity of DON after Ozone Exposure

HepG2 and Hela cells were exposed to DON medium with different concentrations ranging from 0 to 4.0 mg/L and from 0 to 3.0 mg/L, respectively. The MTT assay was used to measure the cell viability and calculate the IC_50_ values. Figure 2 shows the cytotoxic effects of DON on the HepG2 and Hela cells. Inhibition rates of cell viability for the two kinds of cells were significantly increased with the increase of DON concentration (*p* < 0.05), and presented a dose-dependent relation between them. The growth inhibition for HepG2 and Hela cells followed the polynomial model, with correlation coefficients (*R*^2^) of 0.936 and 0.9695, respectively. The IC_50_ values of DON on HepG2 and Hela cells were 2.10 and 1.33 mg/L after 48 h of exposure, respectively, which were calculated based on the corresponding polynomial model.

Based on the IC_50_ values obtained in this study, the initial DON concentrations of the sample without ozone treatment (10.76 ± 0.09 mg/L) were diluted to 2.00 mg/L for HepG2 cells and to 1.50 mg/L for Hela cells with Dulbecco’s modified Eagle’s medium (DMEM) medium. The DON samples exposed to ozone for 5, 10, 15, and 20 min were also diluted to the same concentrations as the ones without ozone treatment. They were used to perform the cytotoxicity assays of DON on HepG2 and Hela cells before and after ozone treatment. Figure 3 demonstrates that ozone exposure significantly reduced the cytotoxicity of DON on HepG2 and Hela cells (*p* < 0.05), and it was dependent on the ozone exposure time. For HepG2 cells, the cell viability evidently increased from 53.95% for the control without ozone treatment to 96.75% for the one treated by ozone for 15 min (*p* < 0.05) (Appendix A). Similarly, for Hela cells, the cell viability rapidly increased from 37.86% for the control without ozone treatment to 91.31% (*p* < 0.05) for the one treated by ozone for 15 min (Appendix A). There were no obvious differences in the cell viability between the negative control (100%, without DON exposure) and the one with DON treatment after ozone exposure for 20 min (*p* > 0.05). 

## 3. Discussion

In 1997, Mckenzie et al. found that ozone can degrade and detoxify common mycotoxins (such as aflatoxins, cyclopiazonic acid, fumonisin B_1_, ochratoxin A, patulin, secalonic acid D, and zearalenone). In recent years, the degradation and detoxification of DON in model solutions and agricultural products have also been explored [11,12,13,14,15,16]. Savi et al. reported that DON levels reduced by 80.67%, 86.60%, and 100% when pericarp grain was exposed to 60 μmol/mol O_3_ for 30, 60, and 120 min, respectively [12]. Similarly, DON contents in contaminated wheat, corn, and bran were reduced by 74.86%, 70.65%, and 76.21% after being exposed to 80 mg/L of saturated aqueous ozone for 10 min [11]. In comparison, the degradation rates of DON in wheat kernels were only 26.40%, 39.16%, and 53.48% after the samples were exposed to 75 mg/L of ozone for 30, 60, and 90 min [13]. The low degradation rates of DON in contaminated wheat (53% and 29%) by ozone treatment were also reported by Wang et al. and Piemontese et al. [15,16]. The large differences in degradation rates of DON in contaminated agricultural products may come from the various ozone detoxification conditions, such as the ozone concentration, ozone state (gaseous or aqueous), ozone flow rate, exposure time, DON levels in contaminated products, water content, and state of treated products [17,18].

Compared with the degradation rates of DON in contaminated agricultural products, they were evidently higher in pure solutions under the same ozone exposure conditions. The degradation rate of DON reached 83% within 7 min when 80 mg/L of gaseous ozone was exposed to 10 mg/L of DON solution, and a first-order kinetic equation described the degradation procedure for DON solution [11]. In our study, results consistent with this were obtained, i.e., the degradation rate of DON was 71.93% when 14.50 mg/L of gaseous ozone was used to treat 10.76 mg/L of DON solution at an 80 mL/min flow rate for 5 min, and reached 97.95% and 100% for 15 min and 20 min, respectively, under the same ozone detoxification conditions. The first-order kinetic model was also used to describe the ozonolysis procedure for DON solution. Li, Guan, and Bian also obtained a 93.6% degradation rate for DON solution (1 μg/mL) treated by gaseous ozone (10 mg/L) for 30 s, while a polynomial equation was used to describe this degradation procedure [19]. These results indicate that DON in a solid state, the DON distribution, and other components in treated agricultural products evidently decrease the degradation rate of DON due to the inadequate contact between them.

IC_50_ is the most widely used and informative measure of a toxin’s toxicity. It indicates how much toxin is needed to inhibit a particular biological or biochemical function by half, thus reflecting the toxicity of this toxin. The MTT assay is one of the most commonly used in vitro methods for preliminarily evaluating the toxicity of mycotoxins and calculating their IC_50_ values due to its advantages of being rapid, convenient, economical, quantitative, and highly reproducible [20]. 

According to the investigations from the reported literature, different IC_50_ values were obtained, depending on the experimental methods, culture conditions, treatment times, and type of cell used [21,22,23,24,25]. For the cytotoxic effects induced by DON in HepG2 cells, IC_50_ values were evaluated in the range from 9.30 μM (~2.75 mg/L) to 2.53 μM (~0.75 mg/L), based on the results from the MTT assays after being exposed to DON (0.625–15 μM) for 24, 48, and 72 h [26]. In this experiment, the IC_50_ values of DON on HepG2 and Hela cells were 2.10 and 1.33 mg/L after 48 h of exposure, respectively, which were in the range of the reported IC_50_ values. 

The toxicity of DON is well-known, with health implications including emesis and anorexia, the alteration of intestinal and immune functions, and reduced absorption of the nutrients, as well as increased susceptibility to infection and chronic diseases. In contrast to DON, very little information exists concerning the toxicity of DON after undergoing ozone detoxification. Animal toxicological experiments have proved that the deleterious effects of DON in contaminated wheats could be greatly reduced by ozone, and ozone itself shows minor toxic effects on animals [16]. Cytotoxicity tests also verified that the toxicity of DON was significantly reduced after being treated by ozone in this study. In addition, DON can inhibit the viabilities of HepG2 and Hela cells, but this does not mean that DON is an anti-cancer toxin. On the contrary, it is a carcinogen according to the animal and cytotoxicity tests [6,27,28].

Overall, these results suggest that ozone can effectively degrade DON in solutions and agricultural products, and the toxic effects induced by DON are significantly reduced after being exposed to ozone for a sufficient amount of time. It is well-known that the toxicity of toxins is closely related to their structures. However, to date, the ozonolysis products of DON are not clear, which has resulted in a limited evaluation on the safety of ozonolysis products of DON. Therefore, in future work, it will be important to identify the structures of the ozonolysis products of DON and to deduce the ozonolysis mechanism of DON, which will help to accurately evaluate the safety of DON after ozone detoxification based on the relation between the structure and toxicity of DON.

## 4. Materials and Methods 

### 4.1. Materials and Cells

The standard deoxynivalenol (purity > 98%) was purchased from Sangon Biotech Co., Ltd. (Shanghai, China). Stock solution of DON (625 mg/L) was prepared with alcohol and maintained at −20 °C in the dark. The working solution of DON was diluted with DMEM medium supplemented with 10% fetal bovine serum (FBS), as well as 1% antibiotic‒antimycotic solution, and the final alcohol content in the culture medium was less than 1% (*v*/*v*).

Dulbecco’s modified Eagle’s medium (DMEM), fetal bovine serum (FBS), and the cell culture flask were provided by Corning Co. (Medford, MA, USA); Trypsin-EDTA (0.25%) solution was purchased from Beyotime biotechnology Co., Ltd (Shanghai, China); 1% antibiotic‒antimycotic solution (100 U/mL penicillin and 100 μg/mL streptomycin), 3-(4,5-Dimethyl-2-thiazolyl)-2,5-diphenyl-2-H-tetrazolium bromide (MTT), phosphate buffer saline (PBS), and 96-well microplates were obtained from Sangon Biotech Co., Ltd. (Shanghai, China); the MTT stock solution (5 mg/mL) was prepared by dissolving 25 mg of MTT powder in 5 mL of PBS solution, and stored at −20 °C; the MTT working solution (0.5 mg/mL) was obtained by diluting 1 mL of stock solution in 9 mL of DMEM medium without FBS; and deionized water was used in all the experiments. 

HepG2 and Hela cells were provided by Fenghui Biotechnology Co., Ltd (Changsha, China). Cells were seeded in 96-well microplates and their viabilities were measured. According to our pre-experimental results, the optimum cell concentration for Hela cells was 5 × 10^4^ cells/well, and 1 × 10^5^ cells/well for HepG2 cells. Cells were allowed to grow and attach to the wells for 20–24 h before treatment with DON.

### 4.2. Determination of DON by HPLC-UV

The HPLC-UV method was used to determine the DON content. DON aqueous solutions were passed through a 0.22 μm filter membrane and injected into the HPLC system (Agilent 1260 with UV detector). The analysis was carried out under the following conditions: the analytical column was Waters XBridge TM (100 × 4.6 mm i.d., 3.5 μm C18 stationary phase); the mobile phase was acetonitrile‒water (6:94, *v*/*v*), with a flow rate of 0.8 mL/min; the oven temperature was set at 35 °C; the wavelength of the UV detector was set at 220 nm; and the injection volume was 20 μL.

### 4.3. Ozone Treatment of DON in Aqueous Solution 

DON in aqueous solution was exposed to gaseous ozone with a reactor (Figure 4). The initial concentration of DON was 10.76 ± 0.09 mg/L, as determined by HPLC-UV. During the process of ozone treatment, ozone was prepared with an ozone generator (Model DJ−Q2020A). Its concentration was monitored in real‒time by an ozone gas analyzer (Model IDEAL−2000), and was controlled at ~15 mg/L (mean 14.50 mg/L). The flow rate of ozone was adjusted to 80 mL/min by a gas flowmeter (Model LZB-4). Before ozone treatment, 10 mL of DON working solution (10.76 ± 0.09 mg/L) was added into the reactor, and then, it was exposed to gaseous ozone under the above-mentioned conditions for times ranging from 0 to 20 min at room temperature. The solution without DON was used as a negative control. All samples were capped and stored immediately in a refrigerator at 4 °C.

### 4.4. Cell Culture and MTT Assay

The HepG2 and Hela cells were grown in the DMEM medium with 10% FBS and 1% antibiotic solution (100 U/mL penicillin and 100 μg/mL streptomycin) at 37 °C in a 5% CO_2_ incubator (Thermo Fisher Scientific, Waltham, MA, USA). The MTT assay was used to determine the IC_50_ values of DON before and after ozone treatment and its cytotoxicity on HepG2 and Hela cells. 

DON solutions with different concentrations (0.0, 1.0, 2.0, 3.0, and 4.0 mg/L for HepG2 cells; 0.0, 1.0, 1.5, 2.0, and 3.0 mg/L for Hela cells) were prepared by diluting the DON (625 mg/L) alcohol solution in the DMEM medium. HepG2 and Hela cells at a concentration of 1 × 10^5^ cells/well and 5 × 10^4^ cells/well were seeded in 96-well flat-bottom plates containing 100 μL of DMEM medium, respectively. After adhering for 20–24 h, HepG2 and Hela cells were treated with different concentrations of DON (100 μL) containing fresh DMEM medium, and then incubated for an additional 48 h. Subsequently, the DMEM medium was removed and 100 μL of MTT solution was added (0.5 mg/mL in DMEM medium without FBS). All the plates were incubated at 37 °C in 5% CO_2_ incubator for 4–6 h, and the formazan crystal was then dissolved in 100 μL of MTT stopping buffer for 16–20 h. Finally, the absorbance was measured at a wavelength of 550 nm using a spectrophotometric microplate reader (Infinite M200 Pro, Tecan, Männedorf, Switzerland). The viabilities of HepG2 and Hela cells were calculated using the following formula:Cell viability (%) = (*A*_Sample_ − *A*_Blank_)/(*A*_Control_ − *A*_Blank_) × 100(1)

The growth inhibition was obtained as the formula: Growth inhibition (%) = 100% − Cell viability (%)(2)

The IC_50_, a concentration of DON at which the growth inhibition rate of cells is at 50%, was calculated by a dose‒response curve using Origin 8.0 (OriginLab Co., USA).

In addition, HepG2 (3 × 10^5^ cells/mL) and Hela (1 × 10^5^ cells/mL) cells were seeded in a 3 mL culture dish, and cultured in the 5% CO_2_ incubator at 37 °C for 24 h. Next, the DMEM medium containing DON, treated by gaseous ozone for different times (0, 5, 10, 15, and 20 min), was added. After being exposed to the above‒mentioned DON solutions for 48 h in the 5% CO_2_ incubator at 37 °C, observations of the densities of HepG2 and Hela cells were made using an inverted microscope (Olympus, Tokyo, Japan).

### 4.5. Statistical Analysis

All experiments were conducted in triplicate, and the results were expressed as the means ± standard deviation (SD). Analysis of variance (ANOVA) was carried out to determine any significant difference (*p* < 0.05) among the applied treatments using the SPSS 18.0 software (IBM, Chicago, IL, USA).

## Figures and Tables

**Figure 1 toxins-11-00639-f001:**
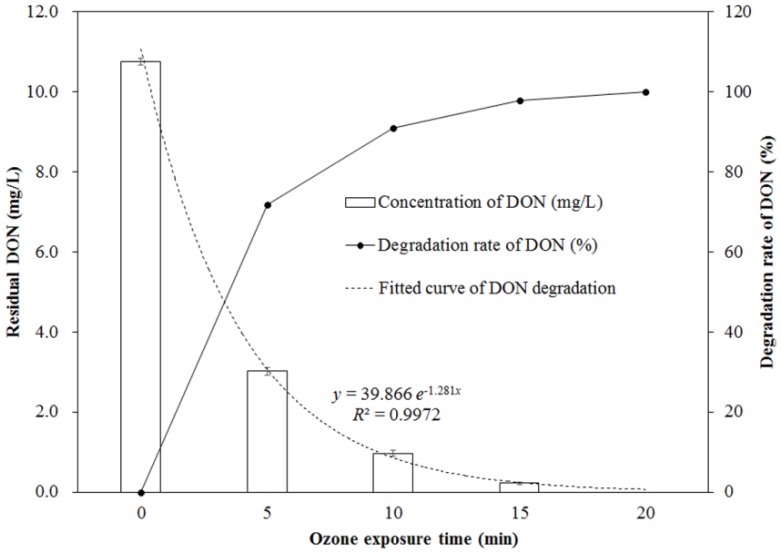
Change of deoxynivalenol (DON) concentration and its degradation rate during ozone exposure.

**Figure 2 toxins-11-00639-f002:**
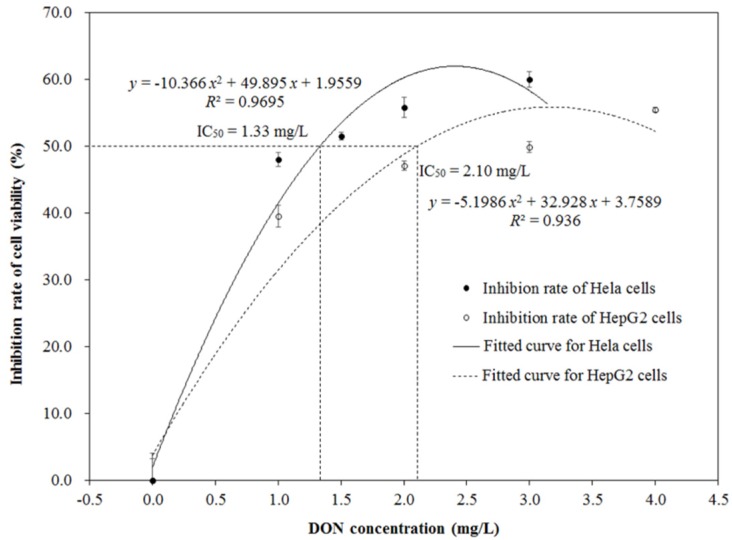
Inhibition rates of cell viability for human hepatic carcinoma (HepG2) and Henrietta Lacks (Hela) cells.

**Figure 3 toxins-11-00639-f003:**
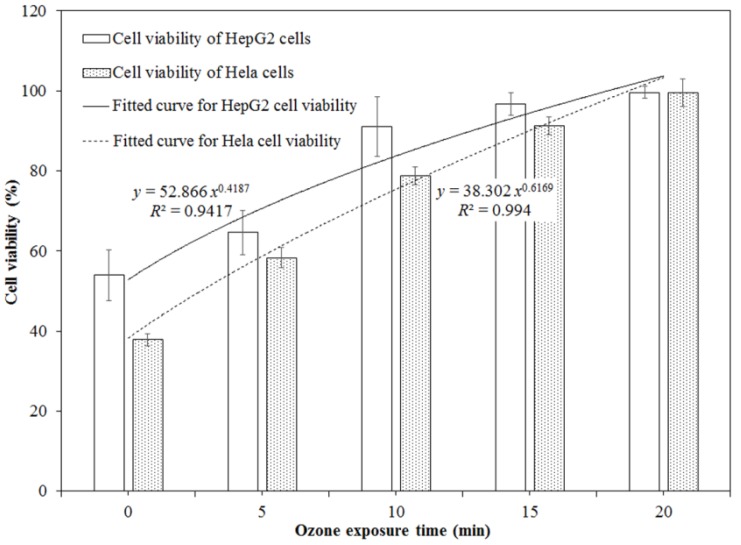
Cell viabilities of HepG2 and Hela cells exposed to DON solution treated by ozone for different exposure times.

**Figure 4 toxins-11-00639-f004:**
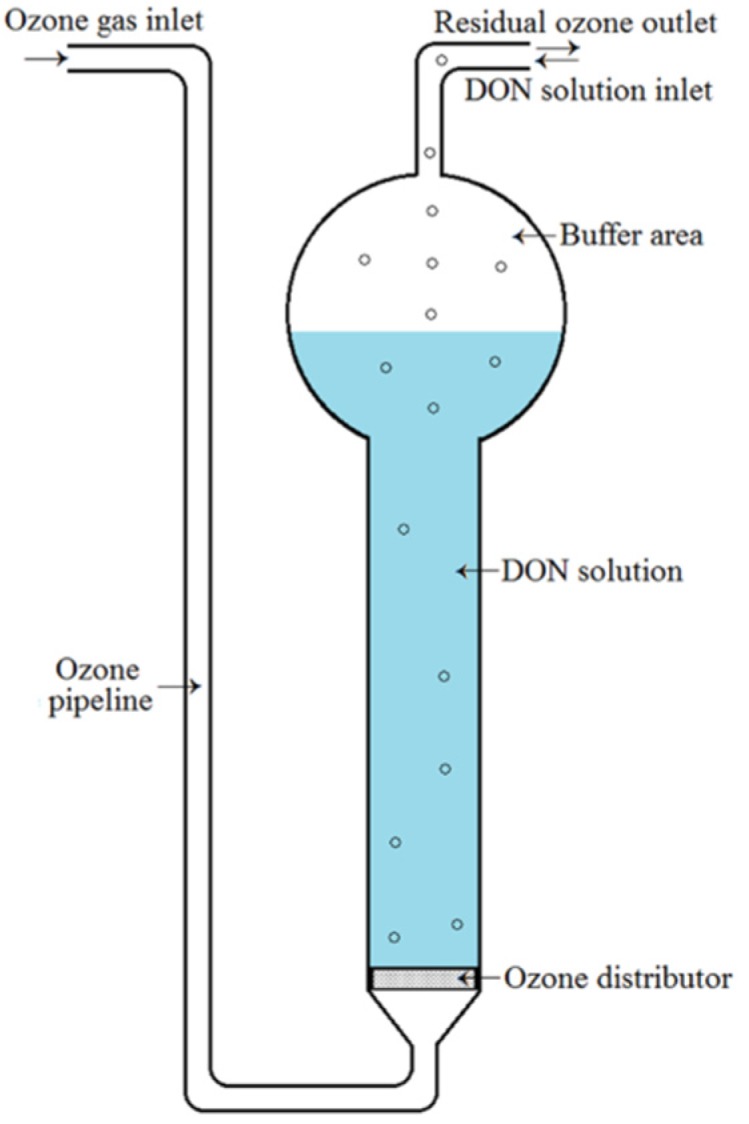
Schematic of the ozone treatment reactor.

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
