# Peer review of "Cytotoxicity of Deoxynivalenol after Being Exposed to Gaseous Ozone"

_toxins, 2019, doi:10.3390/toxins11110639_

Round 1
Reviewer 1 Report
Comments on manuscript: Cytotoxicity of Deoxynivalenol (DON) after being Exposed to Gaseous Ozone.
The author studied the ozonolyzed DON cytotoxicity on cancer cell lines-HepG2 and HeLa.
Following things need to be addressed in MS:
Figure-1- DON degradation increases with increase to ozone exposure time. What major degradation products are formed and what is their role in the current study is missing? Figure 2: Cancer cell lines viability inhibition rate increases with DON concentration. Doesn’t that mean that the DON is an anti-cancer toxin? That in-turn will connect the study with the major degradation product analysis w.r.t. mechanism involved in the inhibition of cell viability. Regarding Figure 5 Micrographs-It would be better to show colony assay formation and analyze the colonies number/size w.r.t. DON concentration or without DON or ozone treated. Also morphology will be much clearer in colony assay formation. Line 24-26 and 142-144, author mentioned that DON contaminates food products, like wheat and other products and ozone treatment will decrease its toxic effects. What about the nutrients or other ingredients in those food molecules after undergoing ozone treatment for DON wipe out? Proofreading is missing throughout the MS: Line 20: Aqueous solution, line 50, which is followed, should be which followed or following, Line 60-same problem, line 83-should be adhered to well not wall.
The paper is technically sound. comments above- will make the MS better.
Reviewer 2 Report
In the presented article the authors evaluate the cytotoxicity of deoxynivalenol (DON) after being exposed to gaseous ozone. Authors carried about careful experiments and got related results, so the manner of scientific work is satisfactory. The manuscript is easy to read but not so well understand in several places. In my opinion, to make it easy to understand for readers, authors should revise the manuscript to improve logical flow, transitions between the sections, and language.
My specific comments are as follows:
Point 1. line 27 page 1, which doses of DON is supposed to be high and causes nausea? Are the doses that authors’ uses in the paper consider to be high or low?
Point 2. Point 2: Have the authors checked the DON by Limulusamebocyte lysate assay? The endotoxin contamination of DON could compromise the results of in vitro.
Point 3. MTT ASSAY
In general, when natural compounds are used in the MTT assay, in the controls, it is used to obtain higher viability values at low concentration of the compound. This might not be related to an increase in the cell growth but to an artifact related to the ability of some of these compounds to interfere with dehydrogenases responsible for the MTT conversion to formazan. The MTT assay for measuring cell viability measure is based on the dehydrogenase activity present in cells. Other methods to measure cell viability should be used to further show that DON induces cell toxicity without affecting any dehydrogenase activity. It is suggested to use a Trypan blue assay or others like with fluorescent dues like Hoechst 33342 for this purpose. Therefore, and as previously suggested, the authors should confirm the viability values with a second method.
Point 4: In the assays with MTT, cell viability should be indicated as cell death (%) instead of inhibition (%) where this percentage is calculated in respect to the control (non-treated cells= 100% of cells alive) (Figure 2).
Point 5: The quality of some figures 4 and 5 are poor.
Point 6: line 118 page 5, change also obtained by have also obtained.
Point 7: line 120-121 page 5, rearrange the sentence is confusing.
Point 8: Determination of DON by HPLC-UV: Based on this, the authors should show how the isolation process was performed and include figures or the chromatograms / spectra obtained during the purification process. These data should be included as supplementary material.
Point 9: line 187 page 6, The MTT stock concentration used in the assay should be indicated.
Point 10: Abbreviations should be avoided in the Abstract.
There are several errors of English grammar and typographical mistakes throughout the manuscript, and these should be corrected.
Reviewer 3 Report
The authors describe the effect of don and its degradations products generated by ozone exposure on two different human cell lines.
The introduction is very short. results which were obtained so far have to be shown in more detail to get an impression of what exactly is already known.
One major concern is the use of the MTT assay to detect cell growth and viability. first, MTT is toxic to eukaryotic cells and in the meantime there are very good alternatives on the market. second, MTT determines the metabolic activity of the cells. so to directly link this measurement to growth rate and viability is not accurate.
Moreover, the authors link viability measurements to IC50 values. if viability is measured then LC50 values should be used. IC50 values are usually applied when any kind of functional inhibition is measured, which is not the case in the present study.
Figure 3 is lacking results of controls - in this case this should be cells that have been treated exactly as exposed cells but without don. Figure 4 and Figure 5 show cell densities. However, the quality of the photograph is really bad so there is no morphological observation possible. also, if don reduced cell numbers it is only obvious that cell densities are also reduced. also the shape of cells changes in relation to the densities and if they are able to grow as a mono layer. There has to be further prove if the morphological differences can be related to the treatment. Microphotographs should always contain scale bars.
The authors should therefore repeat viability or growth rate measurements with appropiate assays, before this article can be considered for publication. also the morphological analyses have to be repeated.
Round 2
Reviewer 1 Report
attachment

Author Response
About the English language and style, moderate English changes requied.Response: Thank you for giving us the good suggestion. English language does present some errors, and we have invited a professional language editing group to help us improve it.
Line 15, adding “there were” before “no significant differences……”.
Line 25, the “,” after “Fusarium culmorum [1]” is replaced by “;”.
Line 37, “……concern by consumers” changes to “……concern of consumers”.
Line 51, the blank space between “which” and “follows a first-order kinetic model” is deleted, i.e. “which follows a first-order kinetic model……”.
Line 101, adding a blank space between the words “and” and “detoxification”.
Line 131, adding a blank space between the words “the” and “investigations”.
Line 144, the word “were” is deleted.
Line 151, adding a blank space between the words “DON” and “and”.
Line 166, the blank space between “……(Shanghai, China);” and “the MTT stock solution……” is deleted.
Line 167, the word “into” changes to “in”; “……stored in -20℃” changes to “……stored at -20℃”.
Line 189-190, “above mentioned” changes to “above-mentioned”.
Line 220, “above mentioned” changes to “above-mentioned”.
Author should mention the name of major products in the revised MS (as data unpublished or with reference if published).Response: In the first response to the comment, we provided the ozonolysis products of DON and their structures (see Figure 1). These results have been submitted to the journal “Toxicon” for reviewing, which have not been published.
Figure 1. Structures of DON and its ozonolysis products
In this study, we explored the cytotoxicity of DON after being exposed to gaseous ozone. Before ozone detoxification, DON is a standard with high purity, and it was converted into a mixture of several ozonolysis products. Based on the analysis of LC-QTOF/MS/MS, we isolated and identified four ozonolysis products of DON. It is very difficult to do the cytotoxicity of the single product due to the lower production. Therefore, the results were the cytotoxicity of residual DON and its ozonolysis products (mixtures). So we don't think it's necessary to name these four products in this study. It does not affect our research objective and results.
In addition, the structures of the four DON degradation products are very complex, and it is very difficult to name them with systematic nomenclature. For convenience purposes, we named them Product 1, 2, 3, and 4, according to their orders of retention times.
It looks author didn’t understand the question and wrote the statement, which is itself contradictory then as why did the author then used this to test toxicity as DON is an exogenous substance to HepG2 cells here?Response: I am sorry that our expression failed to be understood by the reviewer. HepG2 cell is a well-differentiated human liver embryo tumor cell line, which is widely used to perform the cytotoxicity experiments. While the normal hepatocyte cells are rarely used for cytotoxicity tests, because the intrinsic metabolic enzymes in them quickly lost their activity after several splits. HepG2 cell retains complete metabolic enzymes and their activities. The activity of metabolic enzymes for HepG2 cell is very stable and does not decrease with the increase of passage times. So there is no need to add exogenous activation system using HepG2 cell for toxicological test. These characteristics of HepG2 cells are beneficial to toxicological tests and reduce the interference of cell factors to toxic results.
DON as an exogenous substance, which can induce the death of HepG2 cells dues to its toxicity. So it's not contradictory to be used in the MTT assays for cytotoxicity evaluation.
These three paragraphs should be included in revised MS, line 105 before figure 3.Response: Thank you for giving us the good idea. According to the three paragraphs of our reply, we consider that they are more suitable for the discussion. So the first paragraph is placed in Line 145-147. But the second and third paragraphs discussed the relation between the structures and toxicity of DON and its degradation products, and the mechanism of DON inhibiting cell viability, and they're not the focus of this study. So there is no role or sense to put them in the Discussion or Introduction for this study.
5.Colony formation assay or clonogenic assay is especially done for adherent cells. Micrographs provided in Figure 4 and 5 as cell densities are not clearer as well as morphological changes are not visible to differentiate between treated and untreated ones.
Response: The clonogenic assay can be used to evaluate the reproductive viability of adherent cells. While the clonogenic assay is not suitable for all cell lines. Many adherent cell lines do not form colonies. The MTT assay has a number of advantages when compared with a clonogenic assay. It is quick and easy and allows a large number of assays to be carried out in one batch. This is an important consideration when making comparisons between cell lines, between cytotoxic agents, or when evaluating combinations of toxins. No one cytotoxicity assay is ideal, and it is always advisable to support results with those obtained from alternative assays where possible (Plumb, 2004).
Relevant references
Plumb, J.A. (2004) Cell Sensitivity Assays: The MTT Assay. In: Langdon S.P. (eds) Cancer Cell Culture. Methods in Molecular Medicine™, vol 88, pp 165‒169. Humana Press.
HepG2 and Hela cells are monolayer adherent ones, which cannot form colonies like bacteria and yeast. So the clonogenic assay is not suitable for evaluating the vitalities of HepG2 and Hela cells.
The shapes of HepG2 cells are all polygonal and irregular. During the growing process of cells, they are attached to each other by monolayer adherent. When the cell density is large enough, it is really difficult to clearly distinguish the morphological changes. In this study, Figure 4 and 5 mainly show the changes in the numbers of HepG2 and Hela cells (i.e. the changes of cell density), not the morphological changes of cells. So, the figure captions are both changed to “Density changes of HepG2 cells……” and “Density changes of Hela cells……”.

Reviewer 2 Report
In order to improve the quality of your paper within mine suggestion you have done an good job. The only concern that wasn't properly address was regarding the quality of figures 4 and 5 are still poor and the issue that in MTT assay, cell viability should be indicated as cell death (%) instead of inhibition (%) where this percentage is calculated in respect to the control (non-treated cells= 100% of cells alive) (Figure 2).
Author Response
In order to improve the quality of your paper within mine suggestion you have done an good job. The only concern that wasn't properly address was regarding the quality of figures 4 and 5 are still poor and the issue that in MTT assay, cell viability should be indicated as cell death (%) instead of inhibition (%) where this percentage is calculated in respect to the control (non-treated cells= 100% of cells alive) (Figure 2).Response: First of all, thank you for your affirmation to our work. According to your suggestions, we try to improve our manuscript.
The Figure 4 and 5 have improved their quality, and their resolutions have been increased from the original 300 dpi to 600dpi.
About the cell viability, there are different views for different reviewers to express it whether using inhibition rate (%) or death rate (%). MTT assay is quick and easy and allows a large number of assays to be carried out in one batch (Plumb, 2004). In this study, the MTT assay was used to evaluate the cell viabilities of HepG2 and Hela cells, and calculated their IC50 values. IC50, names the half-maximal inhibitory concentrations, indicates how much toxin is needed to inhibit a particular biological or biochemical function by half, thus reflecting the toxicity of this toxin. In order to be consistent with IC50 (the half-maximal inhibitory concentrations), we think it is more appropriate to express cell vitality with inhibition rate (%). In addition, inhibition rate (%) has the meaning of death rate (%).
Relevant references
Plumb, J.A. (2004) Cell Sensitivity Assays: The MTT Assay. In: Langdon S.P. (eds) Cancer Cell Culture. Methods in Molecular Medicine™, vol 88, pp 165‒169. Humana Press.

Reviewer 3 Report
The authors state that giving more details in the introduction would be tedious. However, the novelty of the study seems not to be clear, since it has been stated that
“ozone has been widely used to degrade DON 35 in cereals and cereal-based products due to its strong detoxification capacity, high safety, low cost, 36 and ability to remove DON without leaving toxic residues”. So if it has been shown already that this treatment does not leave toxic residues, why has the present study been performed?
Again, the microscopic pictures do not give any additional value to the study, since viability has already been measured. So why do the authors show pictures of the cell culture, which only confirm what has been shown in the MTT assay?
The manuscript does not include data for a full length original work. It might be better suited for some short communications report.
Author Response
About the English language and style, English language and style are fine/minor spell check required.Response: Thank you for giving us the good suggestion. English language does present some errors, and we have invited a professional language editing group to help us improve it.
Line 15, adding “there were” before “no significant differences……”.
Line 25, the “,” after “Fusarium culmorum [1]” is replaced by “;”.
Line 37, “……concern by consumers” changes to “……concern of consumers”.
Line 51, the blank space between “which” and “follows a first-order kinetic model” is deleted, i.e. “which follows a first-order kinetic model……”.
Line 101, adding a blank space between the words “and” and “detoxification”.
Line 131, adding a blank space between the words “the” and “investigations”.
Line 144, the word “were” is deleted.
Line 151, adding a blank space between the words “DON” and “and”.
Line 166, the blank space between “……(Shanghai, China);” and “the MTT stock solution……” is deleted.
Line 167, the word “into” changes to “in”; “……stored in -20℃” changes to “……stored at -20℃”.
Line 189-190, “above mentioned” changes to “above-mentioned”.
Line 220, “above mentioned” changes to “above-mentioned”.
The authors state that giving more details in the introduction would be tedious. However, the novelty of the study seems not to be clear, since it has been stated that “ozone has been widely used to degrade DON 35 in cereals and cereal-based products due to its strong detoxification capacity, high safety, low cost, 36 and ability to remove DON without leaving toxic residues”. So if it has been shown already that this treatment does not leave toxic residues, why has the present study been performed?Response: As the reviewer said, ozone has been widely used to study the degradation of mycotoxins, including DON. The innovation of this study is not to use ozone to degrade DON, but to evaluate the safety of DON after being exposed to gaseous ozone. In the Introduction of this manuscript, we stated that “ozone has been widely used to degrade DON in cereals and cereal-based products due to its strong detoxification capacity, high safety, low cost, and ability to remove DON without leaving toxic residues”. The “safety” here is to emphasize the safety of ozone. Because the decomposition product of ozone is oxygen, and there are no harmful residues for ozone, except for the toxin itself and its products. Based on the above analysis, we are sorry that our statement in the Introduction does not make it clear to the expert.
The microscopic pictures do not give any additional value to the study, since viability has already been measured. So why do the authors show pictures of the cell culture, which only confirm what has been shown in the MTT assay?Response: In this study, the Figures 4 and 5 are supplement of Figure 3, which can express the changes of cell density (number) more intuitively. Therefore, we think that the Figures 4 and 5 are still necessary.
The manuscript does not include data for a full length original work. It might be better suited for some short communications report.Response: According to the reviewer’s suggestion, we provide the original data for all tests in attachment.

Round 3
Reviewer 3 Report
I observed only marginal changes in the manuscript and therefore refer to my last revision.
Author Response
attachment
